# Multi-User MIMO Downlink Precoding with Dynamic User Selection for Limited Feedback

**DOI:** 10.3390/s25030866

**Published:** 2025-01-31

**Authors:** Mikhail Bakulin, Taoufik Ben Rejeb, Vitaly Kreyndelin, Denis Pankratov, Aleksei Smirnov

**Affiliations:** Radio and Television Department, Moscow Technical University of Communications and Informatics (MTUCI), Moscow 111024, Russia; m.g.bakulin@gmail.com (M.B.); vitkrend@gmail.com (V.K.); dpankr@mail.ru (D.P.); smirnov.al.ed@gmail.com (A.S.)

**Keywords:** B5G, MU-MIMO, downlink precoding, quantized precoding, limited feedback, user selection, massive MIMO

## Abstract

In modern (5G) and future Multi-User (MU) wireless communication systems Beyond 5G (B5G) using Multiple-Input Multiple-Output (MIMO) technology, base stations with a large number of antennas communicate with many mobile stations. This technology is becoming especially relevant in modern multi-user wireless sensor networks in various application scenarios. The problem of organizing an MU mode on the downlink has arisen, which can be solved by precoding at the Base Station (BS) without using additional channel frequency–time resources. In order to utilize an efficient precoding algorithm at the base station, full Channel State Information (CSI) is needed for each mobile station. Transmitting this information for massive MIMO systems normally requires the allocation of high-speed channel resources for the feedback. With limited feedback, reduced information (partial CSI) is used, for example, the codeword from the codebook that is closest to the estimated channel vector (or matrix). Incomplete (or inaccurate) CSI causes interference from the signals, transmitted to neighboring mobile stations, that ultimately results in a decrease in the number of active users served. In this paper, we propose a new downlink precoding approach for MU-MIMO systems that also uses codebooks to reduce the information transmitted over a feedback channel. A key aspect of the proposed approach, in contrast to the existing ones, is the transmission of new, uncorrelated information in each cycle, which allows for accumulating CSI with higher accuracy without increasing the feedback overhead. The proposed approach is most effective in systems with dynamic user selection. In such systems, increasing the accuracy of CSI leads to an increase in the number of active users served, which after a few cycles, can reach a maximum value determined by the number of transmit antennas at the BS side. This approach appears to be promising for addressing the challenges associated with current and future massive MIMO systems, as evidenced by our statistical simulation results. Various methods for extracting and transmitting such uncorrelated information over a feedback channel are considered. In many known publications, the precoder, codebooks, CSI estimation methods and other aspects of CSI transmission over a feedback channel are separately optimized, but a comprehensive approach to jointly solving these problems has not yet been developed. In our paper, we propose to fill this gap by combining a new approach of precoding and CSI estimation with CSI accumulation and transmission over a feedback channel.

## 1. Introduction

MIMO (Multiple-Input Multiple-Output) technology, along with modulation and coding scheme selection, provides a sensible compromise between noise immunity (error correction capability) and data transfer rate in wireless communication systems [1,2]. MIMO systems can, under certain conditions, approach the theoretical limit of channel capacity [3,4,5]. This technology is a highly efficient way of transmitting data and is widely used in modern radio communication systems such as WiFi 5–7 wireless communication systems, LTE, LTE-Advanced, 5G mobile communication systems, etc. It is worth noting the key role of massive MIMO technology in modern mobile communication systems (5G-Advanced), Beyond 5G (B5G) systems and multi-user wireless sensor networks to achieve high-throughput capacity and reliability of data transmission [2,6,7,8].

However, to fully capitalize on the advantages of MIMO systems, especially in a multi-user (MU) scenario, additional signal processing technologies are required, the most important of which is the precoding technology [9,10,11,12,13]. Precoding improves the performance of MIMO systems by using channel information to pre-process the signals transmitted. Precoding is especially effective in Multi-User MIMO (MU-MIMO) systems, as it allows for simultaneous downlink data transmission for users with a small number of receiving antennas using the same frequency resources. Precoders are typically designed in accordance with information theory; so, throughput, capacity, mutual information and other criteria may be used for optimization [2,6,14,15,16].

Precoders that use full MIMO channel state information (full CSI) are well known. References [6,11,12,13,14,17] detail linear precoding approaches that allow for the transmission of users’ signals in the downlink while avoiding interference (i.e., facilitate the full separation of the signals). However, the availability of full CSI without feedback is only possible in TDD mode for low-mobility (stationary) subscribers or in MIMO systems with full-duplex technology [18]. In most other cases, full information is not available on the transmitting side; so, a feedback channel from each user to the base station is used [19,20,21]. With a large number of antennas at the Base Station (BS) and a large number of users in massive MIMO systems, the amount of channel information transmitted over the feedback channel from Mobile Stations (MSs) can be comparable to the total traffic amount, which is a significant limitation. Therefore, partial channel state information (partial CSI) precoding systems that do not require large feedback are of great interest.

For example, in 5G and B5G systems, codebook-based precoding is used to reduce the complexity of feedback and overhead for transmitting CSI [7,9]. With this approach, a Precoding Matrix Indicator (PMI) is transmitted to the BS side using the feedback channel, and the precoder is selected in accordance with a given optimization criterion (e.g., maximum throughput criterion). For 5G-Advanced and future systems with higher throughput requirements, a Discrete Fourier Transform (DFT)-based codebook combined with a low-dimensional search procedure was proposed in [22] for PMI selection to improve the performance, with an acceptable increase in computational complexity. The use of precoding matrix selection on the receiving side and PMI transmission over the feedback channel allows for increasing the communication efficiency for one subscriber but does not solve the problems of the multi-user mode, since the precoding matrix selection is carried out by one user without taking into account the presence of others.

Another valuable approach to limited feedback precoding is based on the use of Grassmannian codebooks [12,21,23,24,25]. In this case, the matrix closest to the actual channel matrix is selected from the Grassmannian codebook on the receiving side, and the index of this matrix is transmitted to the BS. All code matrices received from the MS side are processed at the BS side, and a precoding matrix is formed on this basis. This method can be classified as a multidimensional quantization and, like all quantization methods, provides an estimate of the channel matrix with some error. This quantization error may be reduced by increasing the volume of the codebook (the number of quantized values), but with a large number of antennas at the BS, the required volume of the codebook increases significantly, which complicates the code matrix search process and increases the complexity of the algorithm for selecting a precoding matrix at the MS side. As a result, due to the inaccuracy of channel information, this type of partial CSI precoding is significantly inferior to the full CSI precoding in terms of the number of available active users simultaneously served by the BS. There are methods that partially solve this problem, e.g., in [24], a dual-stage Grassmannian product quantization approach for massive MIMO systems was proposed. This approach is designed to solve the problem of synthesizing large codebooks for the efficient quantization of CSI; however, it only partially takes into account the quality of CSI and does not provide a correction of the MIMO channel matrix estimates, as CSI arrives on the feedback channel from MSs.

The problem of improving the quality (or completeness) of CSI at the base station side in the presence of limited feedback can be solved by using larger codebooks, which is equivalent to increasing the number of quantized values. The problem of codebook synthesis has been solved quite effectively in many recent publications, especially using machine learning and artificial intelligence [26,27], but the large codebook size greatly complicates the codeword selection algorithm. In addition, the strategy of increasing the codebook size is not so efficient in massive MIMO systems, since the number of elements in the channel matrix will increase significantly. The result is insufficient information about the MU-MIMO channel and, as a consequence, a limited number of active users in the MU system on the downlink. It should be noted that the retransmission of the code matrix index to the BS side does not increase CSI, since this is a normal duplication of the previously received information.

The purpose of this work is to improve the accuracy (or completeness) of information about the MU-MIMO channel at the BS side in conditions of limited feedback and use this information to organize communication with the maximum possible number of users. Other known systems use codewords as transmitted CSI [12,23,24]. The difference between this approach and the known ones is that the MS transmits new CSI each time over the feedback channel, uncorrelated with CSI transmitted earlier. This allows for more efficient analysis and accumulation of CSI along with that already obtained.

This allows, after several transmission cycles via the feedback channel, for increasing the accuracy and, as a result, for obtaining almost full CSI and organizing communication on the downlink for the maximum possible number of users. At the same time, the feedback transmission rate does not increase, and CSI accumulates every transmission cycle.

The paper is organized as follows. Section 2 describes the MU-MIMO channel model on the downlink. In Section 3, the basic principles of precoding with full and partial CSI using the MMSE algorithm are considered. Section 4 presents the principle of user selection for communication in MU-MIMO systems on the downlink with partial CSI. Section 5 considers a method for reducing the amount of transmitted CSI via a feedback channel. Quantization using Grassmannian codebooks is considered as such a method. In Section 6, using the material of the previous sections, an approach and an algorithm based on it with cyclic accumulation and channel vector estimates correction in MU-MIMO systems are proposed. Thanks to this algorithm, the completeness of MU-MIMO channel information is constantly increasing, which improves the precoding quality and allows for increasing the number of active users. Section 7 uses statistical simulation to illustrate the effectiveness of the proposed approach and the algorithm that implements it, which leads to a consistent increase in the number of active users served. Finally, Section 8 concludes the paper.

## 2. Downlink MU-MIMO System Model

Let us consider a mathematical model of the multiuser MIMO (MU-MIMO) system, where multiple users or Mobile Stations (MSs) are served simultaneously by one Base Station (BS) on the downlink, and Spatial-Division Multiple-Access (SDMA) technology allows the BS to communicate simultaneously with all MSs through multiple antennas. SDMA is possible with the MIMO technology, and both the BS and the MSs can be equipped with multiple antennas [2,6,7].

Let us introduce the following notations: $N_{tx}$—number of BS transmitting antennas; *k*—index of the MSs; $K_{us}$—number of served MSs; $N_{rx}$—number of receiving antennas at each the MSs; $K_{ch}$—number of spatial streams between BS and MSs; ***H****_k_*—complex downlink MIMO channel matrix between BS and *k*-th MS of dimension $\left(N_{rx}\times N_{tx}\right)$, consisting of complex transmission coefficients between the transmitting antennas of the BS and the receiving antennas of the *k*-th MS; $W_{k}$—precoding matrix at the BS side of dimension $\left(N_{tx}\times K_{ch}\right)$ for *k*-th user’s signal generation; and $W=\left[\begin{array}{cccc} W_{1} & W_{2} & \ldots & W_{K_{us}} \end{array}\right]$—common precoding matrix of dimension $\left(N_{tx}\times K_{ch}K_{us}\right)$, consisting of matrixes $W_{k}$, $k=\bar{1,K_{us}}$. We also assume, for simplicity of further description, that the parameters of all MSs (number of antennas, type of modulation, etc.) are the same. If necessary, all the obtained results can be extended to any combination of MS parameters.

In general, the model of the received signal at the k-th MS receiver input can be written as follows [28]:(1)$$y_{k}=H_{k}W_{k}s_{k}+\sum_{j=1,j\neq k}^{K_{us}}H_{k}W_{j}s_{j}+\eta_{k}=H_{k}Ws+\eta_{k},$$
where $y_{k}$ is the received signal of dimension $\left(N_{rx}\times 1\right)$; $s_{k}$ is a complex vector of transmitted modulated symbols of dimension $\left(K_{ch}\times 1\right)$; s denotes a common vector of information symbols of all users of dimension $\left(K_{ch}K_{us}\times 1\right)$; and $\eta_{k}$ is a complex Gaussian random noise vector in the communication channel of dimension $\left(N_{rx}\times 1\right)$, whose components have zero mean and variance of $2\sigma_{\eta }^{2}$. Equation (1) can be considered as the sum of the useful signals, the signals of neighboring MSs of the multi-user communication system and Additive White Gaussian Noise (AWGN) in the communication channel.

The task of the BS is the simultaneous transmission of signals for $K_{us}$ users selected from the total number $K_{ch}$, using precoding technology in the MIMO channel [2,6,7]. The precoding matrix W, used on the base station side, is calculated after receiving the channel state information from all MSs contained in the common channel matrix H. As a rule, this matrix must ensure the equality of the signals’ powers transmitted to different users, i.e., the following condition must be met:(2)$$\mathrm{trace}\left({W_{k}}^{\prime }W_{k}\right)=1,k=\bar{1,K_{us}},$$
where $A^{\prime }$ denotes the operation of Hermitian conjugation of the matrix A.

## 3. Precoding in an MU-MIMO System with Full and Partial Channel State Information

Let us consider MMSE precoding with fully available CSI using the principle of reciprocity of signal propagation from the BS to the MSs [6,16,28]. To do this, we will solve the inverse problem when the BS receives signals from all MSs. For simplicity of further description and better understanding, we will limit ourselves to considering the case where all MSs have one receiving antenna. In this case, the downlink channel matrix will be a vector $h_{k}$ of dimension $\left(1\times N_{tx}\right)$. This is the most common case and, if necessary, it is quite easy to extend it to any number of receiving antennas, provided that the total number of all MS receiving antennas does not exceed the number of transmitting antennas on the BS side. Let us consider a virtual model of receiving signals from all MSs at the base station:(3)$$y_{BS}=H^{T}s_{MS}+\eta_{BS},$$
where $y_{BS}$ is the vector of the received signals from all users; $H={\left[h_{1}^{T}\ldots h_{K_{us}}^{T}\right]}^{T}$ is the downlink MU-MIMO channel matrix; $H^{T}$ is the MU-MIMO virtual channel matrix for the transmission from MSs to BS in reverse direction; $\eta_{BS}$ is the complex AWGN vector with zero mean and correlation matrix $2\sigma_{\eta }^{2}I_{N_{tx}}$; and $s_{BS}={\left[s_{1}^{BS}\ldots s_{K_{us}}^{BS}\right]}^{T}$ is the vector of the information symbols in the virtual model. Model (3) contains the indexes “BS” and “MS” in order to distinguish the virtual model from model (1), in which the transmission is carried out from the BS to the MSs.

For convenience, we introduce the notation $\overset{˘}{H}≜H^{T}$ and write the MMSE algorithm to find estimates of the users’ symbol vector in the virtual channel:(4)$${\hat{s}}_{MS}=\overset{˘}{G}y_{BS},\;\overset{˘}{G}={\overset{˘}{H}}^{\prime }{\left(2\sigma_{\eta }^{2}I_{N_{tx}}+\overset{˘}{H}{\overset{˘}{H}}^{\prime }\right)}^{-1}={\left(2\sigma_{\eta }^{2}I_{K_{us}}+{\overset{˘}{H}}^{\prime }\overset{˘}{H}\right)}^{-1}{\overset{˘}{H}}^{\prime },$$

In this case, the correlation matrix of the estimation errors will have the following form:(5)$$V=2\sigma_{\eta }^{2}{\left(2\sigma_{\eta }^{2}I_{K_{us}}+{\overset{˘}{H}}^{\prime }\overset{˘}{H}\right)}^{-1}.$$

Taking into account the reciprocity principle, the precoding matrix $\overset{˘}{W}$ is the transposed matrix $\overset{˘}{G}$ of the MMSE estimation algorithm for the virtual channel, [28], i.e.,(6)$$\overset{˘}{W}={\overset{˘}{G}}^{T}={\left({\left(2\sigma_{\eta }^{2}I_{K_{ch}}+{\overset{˘}{H}}^{\prime }\overset{˘}{H}\right)}^{-1}{\overset{˘}{H}}^{\prime }\right)}^{T}={\overset{˘}{H}}^{*}{\left(2\sigma_{\eta }^{2}I_{K_{ch}}+{\overset{˘}{H}}^{T}{\overset{˘}{H}}^{*}\right)}^{-1},$$
where $A^{*}$ denotes the operation of matrix A elements’ complex conjugation. Taking into account the previously introduced matrix notation $\overset{˘}{H}$ in Equation (6), we obtain an expression for calculating the precoding matrix on the BS side without normalization:(7)$$\overset{˘}{W}=H^{\prime }{\left(2\sigma_{\eta }^{2}I_{K_{ch}}+HH^{\prime }\right)}^{-1},$$
where $\overset{˘}{W}=\left[\begin{array}{cccc} {\overset{˘}{w}}_{1} & {\overset{˘}{w}}_{2} & \ldots & {\overset{˘}{w}}_{K_{us}} \end{array}\right]$ is the common precoding matrix of dimension $\left(N_{tx}\times K_{us}\right)$, consisting of unnormalized column vectors $\overset{˘}{w}$ of dimension $\left(N_{tx}\times 1\right)$.

As already noted and expressed in (2), the precoding matrix must ensure equality and constancy of the transmitted signals in the power domain. In this case, the normalization operation can be defined as follows:(8)$$w_{k}=\frac{{\overset{˘}{w}}_{k}}{\sqrt{{{\overset{˘}{w}}_{k}}^{\prime }{\overset{˘}{w}}_{k}}},\;k=\bar{1,K_{us}}.$$

Taking into account the normalization, we obtain the matrix $W=\left[\begin{array}{cccc} w_{1} & w_{2} & \ldots & w_{K_{us}} \end{array}\right]$, which is the common precoding matrix of dimension $\left(N_{tx}\times K_{us}\right)$*,* consisting of vectors $w_{k}$.

Thus, Equations (7) and (8) describe an MMSE precoding algorithm with an accurately known channel matrix H at the BS side. Let us now consider the precoding on the BS side, taking into account the inaccuracy of the channel parameter estimation. As an indicator characterizing the inaccuracy of channel matrix information, we will use the generalized scalar parameter of the total deviation from the actual value, which we will call Channel Quality Information (CQI):(9)$$CQI_{k}≜{\left\|h_{k}-{\hat{h}}_{k}\right\|}^{2}.$$
where ${\hat{h}}_{k}$ is the channel vector estimate.

Let us take into account the inaccuracy of the channel vector estimates ${\hat{h}}_{k}$ on the BS side when forming the precoding matrix. Let us have the estimates ${\hat{h}}_{k}$ of the vectors $h_{k}$, $k=\bar{1,K_{us}}$, whose accuracy is characterized by the corresponding indicator $CQI_{k}$. We approximate the a priori distribution of the original vector $h_{k}$ by a Gaussian distribution with mean value ${\hat{h}}_{k}$ and correlation matrix $V_{k,pr}=v_{k,pr}I_{N_{tx}}$, where $v_{k,pr}$ is a priori variance that will be calculated on the basis of the indicator $CQI_{k}$ using the equation $v_{k,pr}=\frac{CQI_{k}}{N_{tx}}$. Combining AWGN with the noise approximating the inaccuracy of the channel matrix and considering the case of uncorrelated samples of this noise, we obtain the following expression for the unnormalized precoding matrix $\overset{˘}{W}$ (normalization is defined using the expression (8)):(10)$$\overset{˘}{W}={\hat{H}}^{\prime }{\left(\left(2\sigma_{\eta }^{2}+\frac{1}{N_{tx}}\sum_{k=1}^{K_{ch}}CQI_{k}\right)I_{K_{ch}}+\hat{H}{\hat{H}}^{\prime }\right)}^{-1}.$$
where $\hat{H}={\left[{\hat{h}}_{1}^{T}{\hat{h}}_{2}^{T}\ldots {\hat{h}}_{K_{us}}^{T}\right]}^{T}$ is the inaccurately known channel matrix.

The approximate accuracy of the MMSE estimate will be determined by the diagonal elements of the following matrix (correlation matrix of estimation errors):(11)$$V_{PCSI}=\left(2\sigma_{\eta }^{2}+\frac{1}{N_{tx}}\sum_{k=1}^{K_{ch}}CQI_{k}\right){\left(\left(2\sigma_{\eta }^{2}+\frac{1}{N_{tx}}\sum_{k=1}^{K_{ch}}CQI_{k}\right)I_{K_{ch}}+\hat{H}{\hat{H}}^{\prime }\right)}^{-1}$$

The trace of the matrix $V_{PCSI}$ can serve as a general indicator of precoding quality, and each diagonal value of this matrix $v_{PCSI,k}$ will characterize the precoding efficiency for the *k*-th user.

## 4. User Selection MU-MIMO System Taking into Account Partial CSI

The performance of the MU-MIMO system with precoding is very sensitive to the accuracy of the channel vector estimates ${\hat{h}}_{k}$, $k=\bar{1,K_{us}}$. When partial CSI is used on the BS side for precoding, the performance of the MU-MIMO system is significantly degraded. The lack of CSI on the BS side leads to a decrease in the signal-to-noise ratio and to a significant increase in the interference from signals transmitted to other users [4,17,24].

To provide the multi-user mode on the downlink with partial CSI to organize communication, an approach based on user selection from the total number of users waiting for service is used [12,16,25]. Due to inaccurate CSI, precoding does not provide full compensation for interference (signals from other users), especially when receiving signals from MSs with a small number of antennas. This problem can be partially solved by reducing the number of subscribers served, as well as by selecting and combining the subscribers into groups that provide the least influence on each other, taking into account partial channel information.

The problem of user selection cannot be solved only by taking into account the accuracy of channel information (CQI indicator), since this indicator does not take into account the influence of interference. More useful, in this case, is the variance of the MMSE estimation errors $v_{PCSI,k}$ in the virtual channel for one user or the correlation matrix trace for the entire group of users, $\mathrm{trace}\left(V_{PCSI}\right)$, defined by the expression (11).

Let the MIMO system have $K_{us}$ users ready to receive information from the base station. Let us assume that due to the availability of partial information about the channel, communication on the downlink can only be organized for $K_{us,sel}$ users. Therefore, the task is to select $K_{us,sel}$ users out of the total number $K_{us}$, ensuring minimal mutual influence, i.e., the following condition must be met for the selected users:(12)$$V_{PCSI}^{\left(K_{us,sel}\right)}=\underset{{K_{us}}^{︸}\left(\mathrm{trace}\left(P_{NI}{\left(P_{NI}I_{K_{us,sel}}+{\hat{H}}^{\left(K_{us}\right)}{\left({\hat{H}}^{\left(K_{us}\right)}\right)}^{\prime }\right)}^{-1}\right)\right)}{\underbrace{min}}.$$
where $P_{NI}=\left(2\sigma_{\eta }^{2}+\frac{1}{K_{us,sel}}\sum_{k\in K_{us,sel}}CQI_{k}\right)$ is the total power of AWGN and noise, taking into account the uncertainty of channel information, $K_{us}$ indicates the various sets of users’ numbers consisting of $K_{us,sel}$ elements, selected from the total number $K_{us}$, and $K_{us,sel}$ is a set of selected users’ numbers that provide a minimum trace of the matrix, i.e., the optimal combination of users.

Finding the optimal combination of $K_{us,sel}$ users that satisfies condition (12), requires enumerating all combinations from $K_{us}$ of length $K_{us,sel}$, and for each combination it is necessary to compute the inverse matrix of dimension $\left(K_{us,sel}\times K_{us,sel}\right)$. To simplify the user selection procedure, we propose to use an approach without trying all combinations, similar to the approach used in MIMO systems with antenna selection technology [29].

With respect to the considered precoding method, this approach is implemented as follows. We start the selection of combinations with a user with the best quality indicator CQI, then one of all remaining users is matched with this user, for which the minimum trace of matrix $V_{PCSI}^{\left(K_{us,sel}\right)}$ of dimension $\left(2\times 2\right)$ is ensured. Next, for these two users, another user is selected from all the remaining users, in combination with which, the minimum trace of matrix $V_{PCSI}^{\left(K_{us,sel}\right)}$ of dimension $\left(3\times 3\right)$ is ensured. The process of adding the users continues until the specified number $K_{us,sel}$ of users is reached or until some threshold of the matrix $V_{PCSI}^{\left(K_{us,sel}\right)}$ is not reached.

## 5. Channel Matrix Quantization and CSI Reduction That Is Transmitted over the Feedback Channel

The basis for an effective precoding is the availability of CSI on the transmitting side. As already mentioned, in most cases this information is not available at the BS for the downlink transmission; so, a feedback channel from each MS to the BS is needed to transmit the channel information. As before, for simplicity of description, we will consider the case of MSs with one receiving antenna, i.e., the channel on the downlink for one subscriber is described by a complex channel vector $h_{k}$, $k=\bar{1,K_{us}}$, of dimension $\left(N_{tx}\times 1\right)$.

When the quantized values of the channel matrix coefficients are directly transmitting to the BS side, the amount of transmitted information will be equal to $2N_{tx}\log_{2}\left(L_{q}\right)$ bits, where $L_{q}$ is the number of quantization levels; so, with a large number of antennas (massive MIMO), this will require the allocation of a sufficiently large amount of resources [12,23]. Therefore, in order to reduce the amount of information transmitted via the feedback channel, another type of quantization is used, in which a limited set of vectors (codewords) are specified, specially selected and known at both the MS side and the BS side. From the total set of vectors (codebook), the codeword closest to the original vector $h_{k}$ is selected, and only the codeword number is transmitted to the BS. This significantly reduces the amount of information transmitted. Vectors or matrices based on Grassmannians can be used as codebooks (Grassmannian codebooks) [12,21,23,25].

For the 3GPP standards, procedures have been developed for selecting and transmitting via feedback a codeword (the number of one of the codebook vectors) that is closest to the original channel vector according to some criteria [7,9,12,15,22]. Additional information transmitted to the base station may be the scalar parameters coefficient $\alpha_{k}$ and quality indicator $CQI_{k}$, characterizing the correspondence accuracy of this codeword to vector $h_{k}$.

Next, we consider the procedure when the MS selects a codeword from the code book, calculates the complex weighting coefficient $\alpha_{k}$ and the quality indicator $CQI_{k}$, characterizing the correspondence accuracy of this codeword to vector $h_{k}$, and transmits this information along with the selected codeword index $l_{k}$ to the BS.

Let us consider the procedure for selecting the codeword index $l_{k}$ and the estimation of the parameters $\alpha_{k}$ and $CQI_{k}$ in more detail. This procedure can be described by the following equations:(13)$$\left(\alpha_{k},l_{k}\right)=\underset{\alpha \in C,l=\bar{1,N_{v}}}{argmin}\left\{{\left\|h_{k}-\alpha c_{l}\right\|}^{2}\right\},\;CQI_{k}≜{\left\|h_{k}-{\hat{h}}_{k}\right\|}^{2}={\left\|h_{k}-\alpha_{k}c_{l_{k}}\right\|}^{2},$$ where ${\hat{h}}_{k}=\alpha_{k}c_{l_{k}}$ is the quantized channel vector estimate, $c_{l}$ is the *l*-th codeword (normalized vector) from a codebook C of size $N_{v}$, and $CQI_{k}$ is a quality indicator, which is an indicator of the quantization quality of the original vector $h_{k}$ taking into account the complex weighting coefficient $\alpha_{k}$. Thus, the following values are transmitted to the BS side: $\alpha_{k}$, $CQI_{k}$ and the closest codeword index $l_{k}$.

The search for optimal values in (13) is carried out as follows. First, an estimate of the complex weighting coefficient is found:(14)$$\alpha_{l}=h_{k}{c_{l}}^{\prime }{\left(c_{l}{c_{l}}^{\prime }\right)}^{-1}=h_{k}{c_{l}}^{\prime },$$

Substituting (14) in (13), we obtain(15)$$l_{k}=\underset{l=\bar{1,N_{v}}}{argmin}\left\{{\left\|h_{k}-h_{k}{c_{l}}^{\prime }c_{l}\right\|}^{2}\right\}=\underset{l=\bar{1,N_{v}}}{argmin}\left\{\left(h_{k}-h_{k}{c_{l}}^{\prime }c_{l}\right){\left(h_{k}-h_{k}{c_{l}}^{\prime }c_{l}\right)}^{\prime }\right\},$$

Considering that the codewords are normalized, i.e., $c_{l}{c_{l}}^{\prime }=1$, we will finally obtain the following optimal parameters values for precoding:(16)$$\begin{array}{l} l_{k}=\underset{l=\bar{1,N_{v}}}{argmax}\left\{{\left|h_{k}{c_{l}}^{\prime }\right|}^{2}\right\},\\ \alpha_{k}=h_{k}{c_{l_{k}}}^{\prime },\\ CQI_{k}={\left\|h_{k}-\alpha_{k}c_{l_{k}}\right\|}^{2}=\left(h_{k}\left(I_{N_{tx}}-{c_{l_{k}}}^{\prime }c_{l_{k}}\right){h_{k}}^{\prime }\right), \end{array}$$
where $I_{N_{tx}}$ is the identity matrix of dimension $\left(N_{tx}\times N_{tx}\right)$. Obtained with (16), the parameters codeword index $l_{k}$, quantized value of weighting coefficient $\alpha_{k}$, and quality indicator $CQI_{k}$ are transmitted to the BS side. In this case, it is assumed that the BS side has exactly the same set of codebooks as the MS side.

The base station then reconstructs the channel vector estimates ${\hat{h}}_{k}=\alpha_{k}c_{l_{k}}$ for all users $k=\bar{1,K_{us}}$, selects $K_{us,sel}$ users for data transmission and calculates the precoding matrix $W=F\left\{\hat{H}\right\}$, based on the selected precoding criteria $F\left\{\;\right\}$. As such a criterion $F\left\{\;\right\}$, we can use capacity, mutual information, symbol estimation errors in the virtual channel (10) or other criteria [25,28,29].

## 6. Algorithm for Cyclic Accumulation and Correction of Channel Vector Estimates in an MU-MIMO System

The precoding quality depends on the accuracy and amount of CSI that the base station has. In turn, the requirement for high accuracy of CSI is in conflict with the requirement to reduce the amount of information transmitted over the feedback channel. In addition, the requirement to increase quantization accuracy when using a Grassmannian codebook is accompanied by an increase in codebook size and, as a consequence, an increase in the complexity of choosing the appropriate codeword [12,23,24]. For example, if we need to transmit a 16 bit codeword index, when choosing such a word, it will be necessary to perform $2^{16}$ operations of comparison of the original channel vector with the corresponding codeword.

At the same time, CSI from the subscriber to the base station must be updated periodically (cyclically), since the position of the subscribers may change, as well as the number of subscribers participating in communication, etc. To improve the accuracy of CSI at the base station, information cyclically transmitted over the feedback channel can be gradually accumulated and processed. However, if the same information is transmitted (codeword index $l_{k}$, coefficient $\alpha_{k}$ and quality indicator $CQI_{k}$), then the quantization errors on each transmission will be highly correlated, especially when the channel parameters change slowly. Therefore, there will be no significant improvement in the accuracy of partial CSI. It is necessary to ensure that the quantization errors are uncorrelated, as then the obtained information can be accumulated, and the MU-MIMO channel matrix can be restored with higher accuracy.

To reduce the correlation of the quantization errors, we propose to use, instead of a single codebook C of size $N_{v}$, a set of codebooks $C^{\left(n\right)}$, where $n=\bar{1,N_{c}}$, $N_{c}$ is the number of codebooks. Then, before each *n*-th transmission over the feedback channel, the corresponding code book $C^{\left(n\right)}$ is used, and new values of the channel parameters are calculated and transmitted to the BS side:(17)$$\begin{array}{l} l_{k}^{\left(n\right)}=\underset{l=\bar{1,N_{v}}}{argmax}\left\{{\left|h_{k}{c_{l}^{\left(n\right)}}^{\prime }\right|}^{2}\right\},\\ \alpha_{k}^{\left(n\right)}=h_{k}{c_{l_{k}^{\left(n\right)}}^{\left(n\right)}}^{\prime },\\ CQI_{k}^{\left(n\right)}={\left\|h_{k}-\alpha_{k}^{\left(n\right)}c_{l_{k}^{\left(n\right)}}^{\left(n\right)}\right\|}^{2}, \end{array}$$
where $c_{l}^{\left(n\right)}\in C^{\left(n\right)}$ is the l-th codeword from the n-th codebook.

The uncorrelated nature of the quantization errors, in this case, will largely depend on the correlation of the codewords from different books. This creates a certain problem in searching for such books and, in addition, it is necessary to store in the memory of the MSs a set of codebooks ($N_{c}$ codebooks) instead of a single codebook.

Another approach for ensuring uncorrelated quantization errors is to use one codebook, but at each step, it is not the channel vector itself that is quantized, but a modified vector obtained as the result of a special orthogonal transformation of the original channel vector, for example:(18)$$t_{k}^{\left(n\right)}=h_{k}T^{\left(n\right)},\;\left(\alpha_{t,k}^{\left(n\right)},l_{k}^{\left(n\right)}\right)=\underset{\alpha \in C,l=\bar{1,N_{v}}}{argmin}\left\{{\left\|t_{k}^{\left(n\right)}-\alpha c_{l}\right\|}^{2}\right\}.$$
where $T^{\left(n\right)}$ is the matrix of the n-th orthogonal transformation of dimension $\left(N_{tx}\times N_{tx}\right)$, with $N_{tx}$ being the number of antennas at the BS side, and $N_{\nu }$ the size of the original codebook C. As a set of such orthogonal matrices, we propose to use the powers of some initial orthogonal transformation $T_{0}$, i.e., $T^{\left(n\right)}={\left(T_{0}\right)}^{n}$. In this case, to ensure the maximum decorrelation, the original transformation must have the property ${\left(T_{0}\right)}^{n}=I_{N_{tx}}$ only for $n=iN_{tx},i=1,2,3,\ldots$ As a result, similar to (17), we obtain the following expressions:(19)$$l_{k}^{\left(n\right)}=\underset{l=\bar{1,N_{v}}}{argmax}\left\{{\left|t_{k}^{\left(n\right)}{c_{l}}^{\prime }\right|}^{2}\right\},\;\alpha_{t,k}^{\left(n\right)}=t_{k}^{\left(n\right)}{c_{l_{k}^{\left(n\right)}}}^{\prime },\;CQI_{t,k}^{\left(n\right)}={\left\|t_{k}^{\left(n\right)}-\alpha_{t,k}^{\left(n\right)}c_{l_{k}^{\left(n\right)}}\right\|}^{2}.$$

It can be shown that this approach is equivalent to that using the set of codebooks, in which there is one generating codebook, and the others are generated by the orthogonal transformation $T^{\left(n\right)}={\left(T_{0}\right)}^{n}$, i.e.,(20)$$C^{\left(n\right)}=C_{0}{\left(T_{0}\right)}^{n}.$$

The selection of the appropriate codebook $C^{\left(n\right)}$ or the orthogonal transformation $T^{\left(n\right)}$ is carried out at each MS according to an algorithm that is also known on the BS side. The base station at each cycle of receiving partial CSI restores the channel vector estimate of the k-th user. For the first approach, it is calculated simply as follows:(21)$${\hat{h}}_{k}^{\left(n\right)}=\alpha_{k}^{\left(n\right)}c_{l_{k}^{\left(n\right)}}^{\left(n\right)},$$
where the parameters $\alpha_{k}^{\left(n\right)},l_{k}^{\left(n\right)}$ and the quality indicator $CQI_{k}^{\left(n\right)}$ are calculated according to (17) using the set of codebooks $C^{\left(n\right)}n=\bar{1,N_{c}}$ and then transmitted over the feedback channel.

For the second approach, the estimate of the channel vector is calculated through an auxiliary calculation of the vector $t_{k}$ estimate, i.e., we have the following:(22)$${\hat{h}}_{k}^{\left(n\right)}={\hat{t}}_{k}^{\left(n\right)}{T^{\left(n\right)}}^{\prime }\;{\hat{t}}_{k}^{\left(n\right)}=\alpha_{t,k}^{\left(n\right)}c_{l_{k}^{\left(n\right)}},$$
where the parameters $\alpha_{k}^{\left(n\right)},l_{k}^{\left(n\right)}$ are calculated according to expression (19) using one generating codebook $C_{0}$. It can be shown that the quality of the channel vector estimate ${\hat{h}}_{k}^{\left(n\right)}$ will coincide with the quality of the transformed vector estimate ${\hat{t}}_{k}^{\left(n\right)}$:(23)$$\;CQI_{k}={\left\|h_{k}-{\hat{h}}_{k}^{\left(n\right)}\right\|}^{2}={\left\|h_{k}-\alpha_{t,k}^{\left(n\right)}{\hat{t}}_{k}^{\left(n\right)}{\left(T^{\left(n\right)}\right)}^{\prime }\right\|}^{2}=\;{\left\|h_{k}T^{\left(n\right)}-\alpha_{t,k}^{\left(n\right)}{\hat{t}}_{k}^{\left(n\right)}{\left(T^{\left(n\right)}\right)}^{\prime }T^{\left(n\right)}\right\|}^{2}={\left\|t_{k}-{\hat{t}}_{k}^{\left(n\right)}\right\|}^{2}=\;{\left\|t_{k}-\alpha_{t,k}^{\left(n\right)}c_{l_{k}^{\left(n\right)}}\right\|}^{2}=CQI_{t,k}.$$

Equations (17), (21) or (19), and (22) allow for obtaining new channel vector estimates after each new transmission on the feedback channel. Now, the task is to combine all the estimates to improve the accuracy of the final estimate.

We will consider the current channel estimate ${\hat{h}}_{k}^{\left(n\right)}$:(24)$${\hat{h}}_{k}^{\left(n\right)}=h_{k}+\mu_{k}^{\left(n\right)},$$
where $\mu_{k}^{\left(n\right)}$ is the quantization noise, which we approximate by the uncorrelated Gaussian noise with zero mean and variance $r_{k}^{\left(n\right)}$, defined by the quality indicator $CQI_{k}^{\left(n\right)}$, i.e., $\mu_{k}^{\left(n\right)}∼N\left(0,r_{k}^{\left(n\right)}I_{N_{tx}}\right)$, where the noise variance is determined by the expression $r_{k}^{\left(n\right)}=\frac{CQI_{k}^{\left(n\right)}}{N_{tx}}$. Let the vector $h_{k}$ elements be a priori independent random variables with a Gaussian prior distribution with zero mean and variance $v_{pr}$, i.e., $h_{k}∼N\left(0,v_{pr}I_{N_{tx}}\right)$. Using well-known filtering relations (for example, the Kalman filtering algorithm [30,31,32,33]), we obtain the following expressions for the channel vectors estimates of the users and the variances of their estimation errors:(25)$$\;{\tilde{h}}_{k}^{\left(n\right)}={\tilde{h}}_{k}^{\left(n-1\right)}+\kappa_{k}^{\left(n\right)}\left({\hat{h}}_{k}^{\left(n\right)}-{\tilde{h}}_{k}^{\left(n-1\right)}\right),\;v_{k}^{\left(n\right)}=\frac{r_{k}^{\left(n\right)}v_{k}^{\left(n-1\right)}}{\left(r_{k}^{\left(n\right)}+v_{k}^{\left(n-1\right)}\right)},\;\kappa_{k}^{\left(n\right)}=\frac{v_{k}^{\left(n-1\right)}}{\left(r_{k}^{\left(n\right)}+v_{k}^{\left(n-1\right)}\right)},$$
where $\kappa_{k}^{\left(n\right)}$ is the Kalman filter gain for the *n*-th cycle estimation, and $v_{k}^{\left(n\right)}$ is the variance of the channel vector estimation errors. The variance $v_{k}^{\left(n\right)}$ can be used to calculate a new channel estimation quality indicator, as follows:(26)$${\overset{⏜}{CQI}}_{k}^{\left(n\right)}=N_{tx}v_{k}^{\left(n\right)}.$$

The obtained estimates of the channel vectors of all users ${\tilde{h}}_{k}^{\left(n\right)}$ and their quality indicators ${\overset{⏜}{CQI}}_{k}^{\left(n\right)}$ can be used to calculate the correlation matrix of the estimation errors $V_{PCSI}$ (11) and user selection according to algorithm (12).

The considered algorithm (25) of cyclic correction of the channel vector estimates for MU-MIMO system users was obtained for a constant MIMO channel and is intended to reduce the quantization errors when using Grassmannians. However, the proposed algorithm can be modified to filter the parameters of a fluctuating MIMO channel. For this, we consider the first-order model of uncorrelated fading:(27)$$h_{k}^{\left(n\right)}=ah_{k}^{\left(n-1\right)}+\xi_{k}^{\left(n\right)}.$$
where a is a model parameter determined by the nature of fading in the radio channel, and $\xi_{k}^{\left(n\right)}$ is a complex Gaussian noise vector of size $\left(1\times N_{tx}\right)$ with zero mean and correlation matrix $2\sigma_{\xi }^{2}I_{N_{tx}}$.

Using model (27) together with Equation (24), we write estimation algorithm expressions for the channel vectors $h_{k}^{\left(n\right)}$, $k=\bar{1,K_{us}}$ for the *n*-th time point based on the Kalman filter expressions:(28)$${\tilde{h}}_{k}^{\left(n\right)}={\tilde{h}}_{k}^{\left(n-1\right)}+\kappa_{k}^{\left(n\right)}\left({\hat{h}}_{k}^{\left(n\right)}-a{\tilde{h}}_{k}^{\left(n-1\right)}\right),\;v_{k}^{\left(n\right)}=r_{k}^{\left(n\right)}\kappa_{k}^{\left(n\right)},\;\kappa_{k}^{\left(n\right)}=\frac{a^{2}v_{k}^{\left(n-1\right)}+2\sigma_{\xi }^{2}}{\left(r_{k}^{\left(n\right)}+a^{2}v_{k}^{\left(n-1\right)}+2\sigma_{\xi }^{2}\right)}.$$

Block diagrams of the main processing stages on the BS side and on the MS side according to the proposed algorithm (Equations (18), (19), (22) and (28)) are shown in Figure 1 and Figure 2. Figure 2 additionally uses designations for the parameters of the selected users, which are $b_{k,sel}^{\left(i\right)}$, i.e., the *i*-th bit for the *k*-th user, and $\left({\tilde{h}}_{k,sel}^{\left(n\right)},v_{k,sel}^{\left(n\right)}\right)$, i.e., the vector estimate and variance of these estimates for the *k*-th user, where k, in this case, is the number of the selected user, which belongs to the set $k\in K_{sel}$. Blocks designated as “Mx” perform the combining function of symbols streams transmitted and pilot signals. 

Thus, the following precoding Algorithm 1 is proposed for the MU-MIMO system, which can be written in steps on the MS side and BS side (all these steps are repeated at each n-th cycle of receiving new CSI from the MS side).

**Algorithm 1** Algorithm for the cyclic accumulation and correction of channel vectors estimates in the MU-MIMO system.
**MS side**
Channel vector $h_{k}$ estimation using pilot signals.Channel vector transformation $t_{k}^{\left(n\right)}=h_{k}T^{\left(n\right)}$.Quantization of the transformed vector and calculation of the parameters $\alpha_{t,k}^{\left(n\right)},l_{k}^{\left(n\right)}$ and the quality of the quantization parameter $CQI_{t,k}$ according to (18).Transmission of the parameter values $\alpha_{t,k}^{\left(n\right)},l_{k}^{\left(n\right)}$ and $CQI_{t,k}$ via the feedback channel.

**BS side**
The base station, in accordance with (22), makes a rough estimate ${\hat{h}}_{k}^{\left(n\right)}$ of the channel vectors $h_{k}$ based on the current CSI received via the feedback channel from the MSs.Using the previous estimates ${\tilde{h}}_{k}^{\left(n\right)}$ (from the previous cycles), the BS performs a correction of the channel vector estimates for all MSs using the Kalman filter (28).The estimates of the channel vectors ${\tilde{h}}_{k}^{\left(n\right)}$ and the variance $v_{k}^{\left(n\right)}$ of their estimation error obtained as a result of filtering (cyclic correction and accu-mulation) are used to select users (according to condition (12)) and calculate the pre-coding matrix (10) taking into account the normalization (8).


The main feature of this algorithm is that the MSs each time select the most suitable codewords from different codebooks. This new information is transmitted to the BS side. Due to the uncorrelated nature of this information, its accumulation becomes possible, and the accuracy of estimating the MU-MIMO channel matrix increases. As a result, this allows us to get closer to an algorithm with a precisely known channel matrix, i.e., full channel information (full CSI), even with a limited amount of CSI transmitted over the feedback channel. This approach can be used not only in combination with the precoding algorithm based on the MMSE criterion, but also in combination with other high-performance precoding algorithms [34,35,36].

## 7. Simulation Results

To confirm the effectiveness of the proposed precoding algorithm with quantization, accumulation and correction of CSI and user selection in the MU-MIMO system (Equations (18), (19), (22), (28)), a computer statistical simulation was performed in the MATLAB system. The performance comparison was based on noise immunity and system capacity. The simulation parameters are listed in Table 1. Moreover, we considered a scenario where the maximum number of users was equal to the number of spatial channels, i.e., $K_{us}=K_{ch}$, and each user had one receiving antenna. The noise immunity characteristics were investigated for a fixed number of users equal to the maximum possible number $\left(K_{us,sel}=K_{ch}\right)$, with a different number of transmission cycles over a feedback channel. The object for comparison was a precoding algorithm based on Grassmanian codebooks [12,23], which coincides with the algorithm described in this article, i.e., (18)–(19), (22) and (28), but does not use orthogonal transformation before choosing the codeword. This algorithm corresponds to the case of the proposed precoding approach for *n* = 1 using a Grassmannian codebook of the same size.

As the base orthogonal matrix, the $T_{0}$ selected permutation matrix was used for transformation to ensure the maximum repetition period, i.e., ${\left(T_{0}\right)}^{n}=I_{N_{tx}}$, only for $n=iN_{tx},i=1,2,3,\ldots$

As a result of the simulation, curves were obtained (Figure 3), showing the dependence of the Frame Error Rate (FER) with channel coding on the Signal-to-Noise Ratio (SNR) per bit for different numbers of CSI transmission cycles on the feedback channel for the number of users served equal to the maximum number ($K_{us,sel}=32$). Figure 4 shows the FER dependence on the number of cycles for different values of the SNR per bit. A simple precoding algorithm using only Grassmannian quantization (“Grassm. only”) with a size of 1024 codewords (10 bits in feedback) does not work for this simulation case (32 antennas at the BS side and 32 MSs with one antenna each), since such quantization does not give enough CSI to provide efficient precoding for all 32 users. For such a number of users, it is necessary to use a Grassmannian with a significantly larger number of codewords, which is unacceptable due to the high computational complexity. Its characteristics coincide with the characteristics of the proposed algorithm for *n* = 1, and each subsequent transmission over feedback channel does not improve the accuracy of CSI.

From the above Figures, it can be seen that, starting from the forth cycle, for the proposed algorithm, each new cycle of quantized CSI transmission from MSs to BS leads to a gradual improvement in the noise immunity characteristics of the MU-MIMO system, which confirms the effectiveness of the proposed precoding algorithm with CSI accumulation and correction of the MIMO channel matrix estimates. The above Figures show that for all 32 users, the FER = 0.01 level is achieved at 16th cycle with SNR = 0 dB, FER = 0.001 is achieved at 20th cycle with SNR = 2 dB, and when the number of cycles increases to 32, the energy efficiency increases by 4 dB for both FER values.

The “Grassm. only” curve shows that this approach does not allow for increasing the amount of CSI, since the same quantized information is transmitted over the feedback channel, i.e., no CSI accumulation occurs. As a result, the number of users served remains at the same level and does not increase. The purpose of these Figures, where a comparison with the “Grassm. only” curve occurs, is to show that at each CSI transmission cycle, the proposed approach allows for accumulating CSI and thereby achieving the maximum number of MU-MIMO users on the downlink.

These simulation results confirm the effectiveness of the proposed precoding algorithm with CSI accumulation and correction of the MIMO channel matrix estimates for a fixed number of active (served) users. However, of greater interest is the use of this approach in a dynamic user selection scenario, where upon receipt of each new CSI transmission over the feedback channel, the channel matrix estimates are refined, and the number of users selected to organize communication in the downlink increases. For this purpose, a preliminary simulation of the MU-MIMO system was carried out with different numbers of selected users. For each fixed number of users, the minimum number of transmission cycles on the reverse channel was determined to ensure the same FER = 0.01 with approximately the same SNR. Moreover, the approximate SNR value for the maximum number of selected users $K_{us,sel}=32$ was chosen.

Figure 5 shows the dependence of the selected user number $K_{us,sel}$ on the cycle number n (feedback channel transmission) for the proposed algorithm (curve marked “Proposed Precoding (Ort.Tr.+Quant.+Filt.)”), i.e., with orthogonal transformation of the channel vector, quantization and additional filtering of the quantized channel estimates. It also shows the dependence for the known precoding algorithm with a single Grassmannian codebook (“Precoding with Grassmannian only”). As it is known [12,23,24,26,27], the main advantage of the Grassmannian codebooks is the effective CSI amount compression transmitted over the feedback channel. But the completeness of CSI and, accordingly, the quality of channel matrix reconstruction at the BS side strongly depend on the size of the codebook and the size of the channel matrix. For this system configuration, with a codebook size of 1024 complex codewords (each codeword consists of 32 symbols), it is impossible to provide communication on the downlink for more than 4 users out of 32 users. In addition, the same codebook and the same channel vector (or matrix) are used to select the codeword; so, the same codeword is selected. The amount of information about the communication channel at the BS side does not increase, and accordingly, the number of active users in the MU-MIMO system will also remain constant.

Here, we can see that compared to the single-codebook precoding algorithm, the proposed approach allows for consistently increasing the number of active users from 4 to a maximum number of 32 in 18 cycles.

Figure 6 shows the dependence of the FER on the SNR per bit for different number of cycles and different number of selected users. These curves show that the noise immunity characteristics change insignificantly with a simultaneous increase in the number of cycles and the number of users, and the spread of the noise immunity characteristics is ~1 dB at the FER = 0.01 (1%) level. Moreover, compared to simple precoding with one codebook (the characteristic coincides with the curve for *n* = 1), there is an eightfold increase in the number of users for 18 CSI transmission cycles on the feedback channel.

It should be noted that the above Figures not only confirm the efficiency of the proposed approach, but also generally demonstrate the advantages of the precoding technology, which ensures high-quality communication with sufficiently low signal-to-noise ratio values. This can be explained by the fact that in addition to reducing interference from signals from neighboring users, two more effects are observed when using precoding with 32 transmitting antennas. First, signals with independent fading conditions from each transmitting antenna are combined together, resulting in a diversity reception effect. Second, the parameters of the precoding matrix are calculated in such a way as to not only reduce the influence of the signals from other users, but also increase the power of the received signal from the desired user due to the optimal combination of the signals from different antennas.

## 8. Conclusions

Based on the simulation results, it can be concluded that the proposed precoding algorithm with quantization, CSI accumulation, CSI correction and user selection in a MU-MIMO system (expressions (18), (19), (22), (28)), allows for significantly increasing the number of active users in the MU-MIMO system with a large number of antennas on the downlink. The increase in the number of active users occurs gradually, as the information accumulates, and does not affect the quality of communication of already selected users. This approach is most effective in multi-user communication systems that use limited feedback to transmit CSI to the BS side (5G and B5G) [26,27,35]. By using special orthogonal transformations before quantization, CSI at the BS side can be incrementally updated and improved without increasing the capacity of the feedback channel. As the quality of the MIMO channel estimates improves, the number of active users supported by the MU-MIMO system grows and reaches the maximum number of users determined by the spatial resources allocated in the massive MIMO system.

The proposed algorithm fits well with the algorithm for correcting the estimates of the wireless communication channel, since the newly arriving channel state information through the feedback channel to the BS allows, at the same time, for refining the insufficiently accurate channel estimates previously obtained, as well as for correcting them taking into account the change in the channel state for each user. Thus, the proposed approach for organizing communication with precoding, compared with the conventional precoding algorithm based on the use of codebooks, allows for a gradual increase in the number of active users and the achievement of precoding efficiency in MU-MIMO systems, without increasing the transmission rate on the feedback channel.

The approach to extracting uncorrelated CSI described in the paper is well compatible with various methods of optimizing code sequences for CSI transmission using limited feedback [26,27]. Its use is not limited to MMSE precoding, as it can also be combined with various precoding and beamforming methods [26,35,37].

In future work, we plan to continue the study of the precoding approach with dynamic user selection, especially of the influence of the total number of users requesting service in massive MIMO systems on the speed of reaching the maximum number of active users, and also to consider scenarios where the total number of users in the MU-MIMO system exceeds the number of available spatial channels.

## Figures and Tables

**Figure 1 sensors-25-00866-f001:**
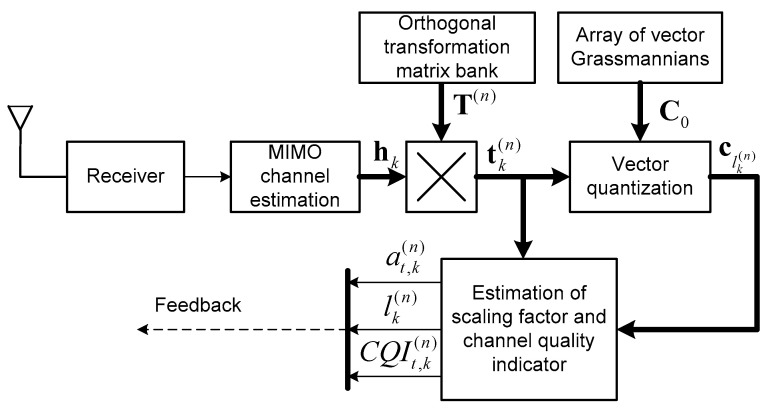
Block diagram of the k-th MS with MIMO channel estimation, quantization and transmission of the precoding parameters using feedback to the BS (for the n-th time point).

**Figure 2 sensors-25-00866-f002:**
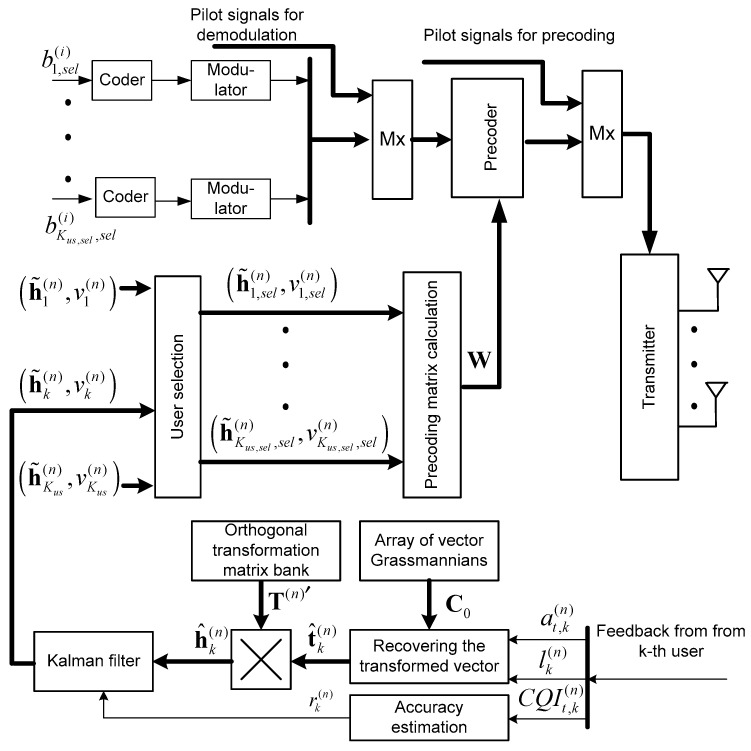
Block diagram of the BS with the precoding algorithm using CSI accumulation and filtration and dynamic user selection (for the *n*-th time point).

**Figure 3 sensors-25-00866-f003:**
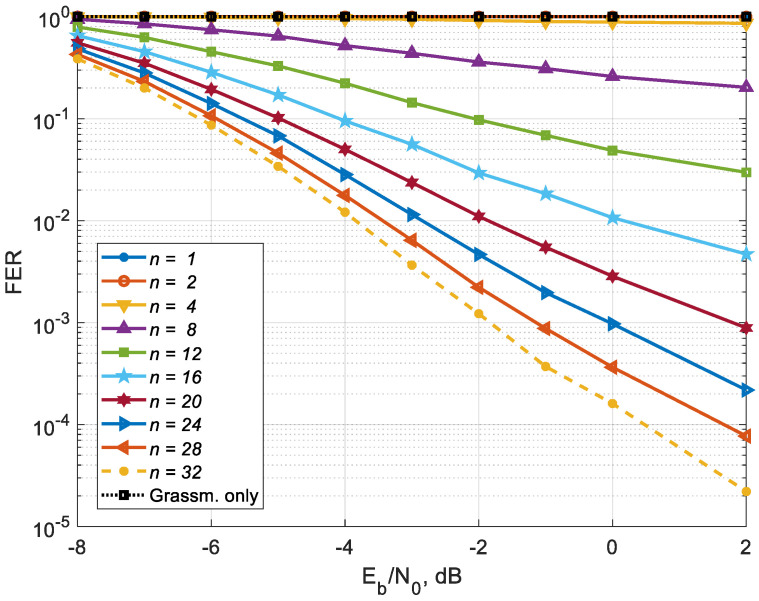
FER dependence on the SNR per bit for different numbers of CSI transmission cycles over the feedback channel.

**Figure 4 sensors-25-00866-f004:**
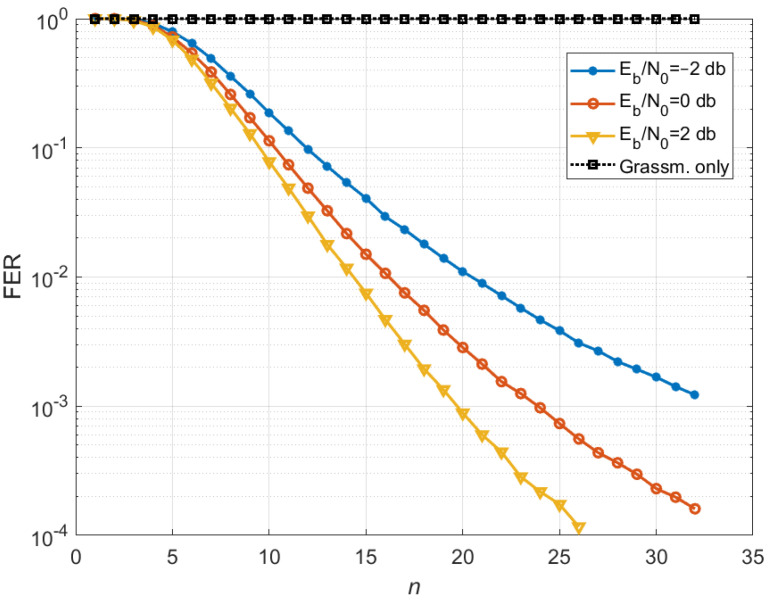
FER dependence on the number of CSI transmission cycles on the feedback channel for different SNRs.

**Figure 5 sensors-25-00866-f005:**
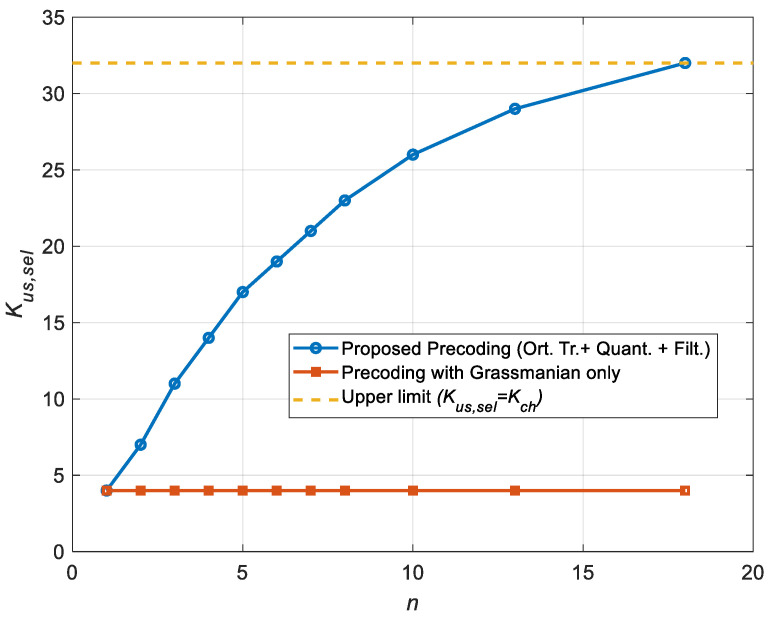
The dependence of the selected (active) user number on the number of cycles (transmission cycle number *n*) for the proposed precoding approach and the known single-codebook precoding algorithm.

**Figure 6 sensors-25-00866-f006:**
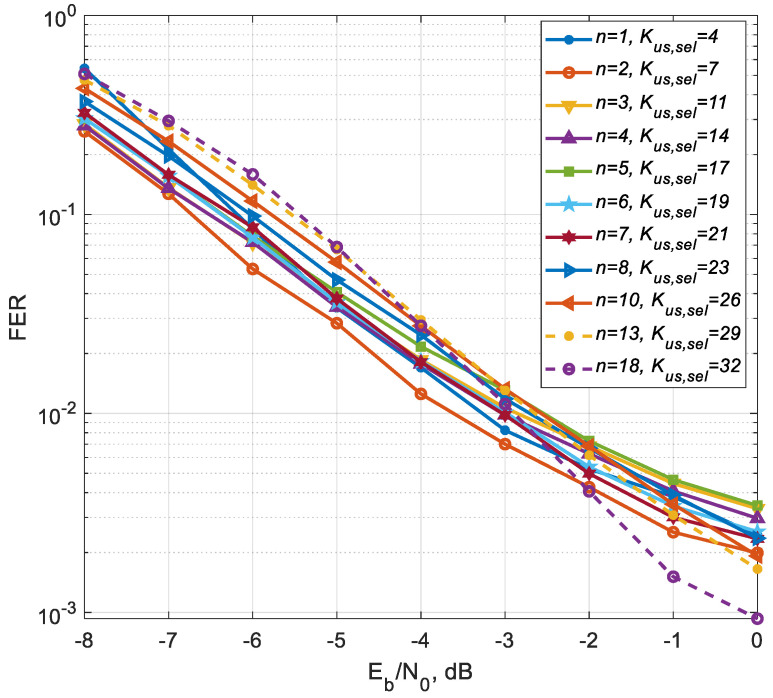
FER dependence on SNR per bit for different numbers of CSI transmission cycles and selected users.

**Table 1 sensors-25-00866-t001:** Computer simulation parameters.

Parameters	Values
Number of base station transmitting antennas	32
Maximum number of served MSs (active users)	32
Grassmannian codebook size	1024
Modulation type	QPSK
Code rate (turbo coding)	1/3
Channel model	MU-MIMO channel, slowly independent Rayleigh fading, NLOS
Demodulator on MS side	MMSE
Precoder on BS side	MMSE

## Data Availability

The data presented in this study are available on request from the corresponding author due to participant privacy restrictions.

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
