# Peer review of "Multi-User MIMO Downlink Precoding with Dynamic User Selection for Limited Feedback"

_sensors, 2025, doi:10.3390/s25030866_

Round 1
Reviewer 1 Report
Comments and Suggestions for Authors
This work proposes a new downlink precoding approach with dynamic users selection for MU-MIMO systems, which also uses codebooks to reduce the information transmitted over feedback channel, but unlike in the existing approaches, here new information uncorrelated with the previous one is transmitted on each new transmission cycle. This paper is timely but the following issues are suggested to address.
1. The abstract is suggested to further summarize for better readability.
2. The authors may explain why the case of Grassm, only, performs so worse?
3. The authors may review the recent progress in MIMO systems in the introduction, such as Channel Estimation for Movable-Antenna MIMO Systems via Tensor Decomposition, Energy-Efficient Beamforming for Downlink Multi-User Systems with Dynamic Metasurface Antennas, where the overhead and downlink precoding performance can be effectively improved.
4. Why the proposed method can work in a such low Eb/No regime?
Author Response
Point-by-point response to the Reviewer's comments
Sensors Journal
Manuscript ID: sensors-3431905
Multi-User MIMO downlink precoding with dynamic users selection for limited feedback
Dear Editor and Reviewers,
Thank You very much for your time and valuable and constructive comments on our article. We took into account all Your comments and hope the quality of the paper was enhanced. In the following text we give our answers to the comments received point-by-point. For the convenience each response was marked in yellow color and we also highlighted the updates in yellow in the revised version of the article.
Corresponding authors,
Dr. Mikhail Bakulin, Dr. Taoufik Ben Rejeb, Prof. Dr. Vitaly Kreyndelin, Dr. Denis Pankratov and Dr. Aleksei Smirnov,
January 20, 2025
REVIEWER # 1
Open Review
(x) I would not like to sign my review report
( ) I would like to sign my review report
Quality of English Language
(x) The quality of English does not limit my understanding of the research.
( ) The English could be improved to more clearly express the research.
Comments and Suggestions for Authors
This work proposes a new downlink precoding approach with dynamic users selection for MU-MIMO systems, which also uses codebooks to reduce the information transmitted over feedback channel, but unlike in the existing approaches, here new information uncorrelated with the previous one is transmitted on each new transmission cycle. This paper is timely but the following issues are suggested to address.
- The abstract is suggested to further summarize for better readability.
We thank the Reviewer for this valuable remark. The abstract has been revised and added for better readability and understanding of the article’s key points – please see lines 9-44.
- The authors may explain why the case of Grassm, only, performs so worse?
We thank the Reviewer for pointing out this issue. The “Grassm. only” curve shows that this approach does not allow increasing the amount of CSI information, since the same quantized information is transmitted over the feedback channel, i.e., no information accumulation occurs. As a result, the number of supported users remains at the same level and does not increase. The purpose of these Figures, where a comparison with the “Grassm. only” curve occurs, is to show that at each CSI transmission cycle, the proposed approach allows accumulating information about the channel and thereby achieving the maximum number of users in the downlink of MU-MIMO system. Please see lines 477-484.
We also add the following explanation on lines 453-459:
A simple precoding algorithm using only Grassmannian quantization (“Grassm. only”) with a volume of 1024 code words (10 bits in the feedback) does not work for this simulation case with 32 antennas at the base station and 32 mobile stations with one antenna each, since with such quantization there is not enough CSI information to provide efficient precoding for all 32 users. For such a number of users, it is necessary to use a Grassmannian with a significantly larger number of codewords, which is unacceptable due to the high computational complexity.
- The authors may review the recent progress in MIMO systems in the introduction, such as Channel Estimation for Movable-Antenna MIMO Systems via Tensor Decomposition, Energy-Efficient Beamforming for Downlink Multi-User Systems with Dynamic Metasurface Antennas, where the overhead and downlink precoding performance can be effectively improved.
Thank You for this remark. The proposed approach based on the transmission of uncorrelated information about the MU-MIMO channel can be used not only with the described MMSE precoding algorithm, but also in combination with other highly efficient precoding algorithms. This is indicated in the article and we added more explanation (please see the lines 412-421). The following approaches related to channel estimation for movable-antenna MIMO systems, efficient beamforming approach for downlink MU systems and metasurface antennas have been reviewed and added to the manuscript (please see the references [35, 36, 37], highlighted in yellow):
- R. Zhang et al., "Channel Estimation for Movable-Antenna MIMO Systems via Tensor Decomposition," in IEEE Wireless Communications Letters, vol. 13, no. 11, pp. 3089-3093, Nov. 2024, doi: 10.1109/LWC.2024.3450592;
- G. Chen, R. Zhang, H. Zhang, C. Miao, Y. Ma and W. Wu, "Energy-Efficient Beamforming for Downlink Multi-User Systems with Dynamic Metasurface Antennas," in IEEE Communications Letters, doi:10.1109/LCOMM.2024.3513554;
- J. -C. Chen and C. -H. Hsu, "Beamforming Design for Dynamic Metasurface Antennas-Based Massive Multiuser MISO Downlink Systems," in IEEE Open Journal of the Communications Society, vol. 5, pp. 1387-1398, 2024, doi:10.1109/OJCOMS.2024.3367984.
- Why the proposed method can work in a such low Eb/No regime?
We appreciate this Reviewer's remark. The Figures are plotted on the base of the average signal-to-noise ratio in the channel, without taking into account precoding, for a channel with independent Rayleigh fading. When using precoding for 32 transmitting antennas, two effects are observed. First, signals with independent fading from each transmitting antenna are added together, which leads to the effect of diversity reception. Second, the parameters of the precoding matrix are calculated in such a way that not only the influence of signals from other subscribers is reduced, but also the power of the received signal from one user is increased due to the optimal addition of signals emitted by different antennas. All these factors allow to obtain significant energy gain. We have added the corresponding explanation in the lines 530-540.
Thank You very much for these valuable comments and issues!
We are looking forward to Your reply.
Kind Regards, authors.

Reviewer 2 Report
Comments and Suggestions for Authors
Dear Authors,
Thank you for the opportunity to review your paper, “Multi-User MIMO Downlink Precoding with Dynamic User Selection for Limited Feedback.”
Your work introduces a novel downlink precoding approach for MU-MIMO systems, focusing on dynamic user selection while leveraging codebooks to reduce the information transmitted over the feedback channel. A key aspect of your method is the transmission of new, uncorrelated information on each cycle, enabling the accumulation of channel state information (CSI) with greater accuracy without increasing feedback overhead. This approach appears promising in addressing challenges associated with modern and future Massive MIMO systems, as highlighted by your statistical simulations.
The paper is well-structured and addresses an important topic, bringing novelty to the specific case of dynamic user selection for limited feedback in MU-MIMO systems. However, I would like to offer some suggestions to enhance the clarity and impact of the work:
1. Introduction and Research Gap:
While the Introduction is well-organized, the research gap could be articulated more explicitly. It would be beneficial to clarify how the proposed method differs significantly from prior approaches and to provide a stronger motivation for the study. Similarly, a clearer articulation of the paper’s objectives in the context of this gap would enhance the reader's understanding.
2. Methodology:
The methodology section would benefit from additional detail. For instance, a more comprehensive explanation of the algorithm and its theoretical underpinnings could help readers better grasp its contributions and the innovations it brings.
3. Literature Review and References:
While the paper presents an interesting concept, much of the referenced literature appears outdated, with most references exceeding 10 years in age. Incorporating more recent studies, particularly those related to 5G and B5G systems, would strengthen the relevance of the work and provide a more robust foundation for your claims.
4. Conclusion:
The conclusion could be more precise and aligned with the presented results. Currently, statements such as “The proposed precoding algorithm with quantization can be used for dynamic downlink user selection in MU-MIMO systems with a large number of antennas (5G and B5G)” feel somewhat ambiguous without clearer ties to your findings. Reiterating how the results validate the approach and explicitly connecting the outcomes to practical applications would make the conclusion more compelling.
These improvements would enhance the clarity, impact, and academic rigor of your paper.
Thank you for your contribution to this important area of research. I look forward to seeing the revised manuscript.
Best regards,
The reviewer

Author Response
Point-by-point response to the Reviewer's comments
Sensors Journal
Manuscript ID: sensors-3431905
Multi-User MIMO downlink precoding with dynamic users selection for limited feedback
Dear Editor and Reviewers,
Thank You very much for your time and valuable and constructive comments on our article. We took into account all Your comments and hope the quality of the paper was enhanced. In the following text we give our answers to the comments received point-by-point. For the convenience each response was marked in yellow color and we also highlighted the updates in yellow in the revised version of the article.
Corresponding authors,
Dr. Mikhail Bakulin, Dr. Taoufik Ben Rejeb, Prof. Dr. Vitaly Kreyndelin, Dr. Denis Pankratov and Dr. Aleksei Smirnov,
January 20, 2025
REVIEWER # 2
Open Review
(x) I would not like to sign my review report
( ) I would like to sign my review report
Quality of English Language
(x) The quality of English does not limit my understanding of the research.
( ) The English could be improved to more clearly express the research.
|
|
Dear Authors,
Thank you for the opportunity to review your paper, “Multi-User MIMO Downlink Precoding with Dynamic User Selection for Limited Feedback.”
Your work introduces a novel downlink precoding approach for MU-MIMO systems, focusing on dynamic user selection while leveraging codebooks to reduce the information transmitted over the feedback channel. A key aspect of your method is the transmission of new, uncorrelated information on each cycle, enabling the accumulation of channel state information (CSI) with greater accuracy without increasing feedback overhead. This approach appears promising in addressing challenges associated with modern and future Massive MIMO systems, as highlighted by your statistical simulations.
The paper is well-structured and addresses an important topic, bringing novelty to the specific case of dynamic user selection for limited feedback in MU-MIMO systems. However, I would like to offer some suggestions to enhance the clarity and impact of the work:
We kindly thank the Reviewer for the great work in reviewing the paper and for the useful and valuable comments which we hope helped us to improve the manuscript. For convenience, under each point we provide our comments on the comments and the corresponding corrections made to the article, presented in color text.
- Introduction and Research Gap:
While the Introduction is well-organized, the research gap could be articulated more explicitly. It would be beneficial to clarify how the proposed method differs significantly from prior approaches and to provide a stronger motivation for the study. Similarly, a clearer articulation of the paper’s objectives in the context of this gap would enhance the reader's understanding.
Thank You for the valuable suggestions. The Introduction section has been corrected, the purpose of the paper and a simple explanation of how the proposed approach differs from known ones in the context of the research gap have been added. Please see the lines 110-119 and 122-129 of the revised version of the manuscript. The Abstract also have been modified according to Your suggestions.
- Methodology:
The methodology section would benefit from additional detail. For instance, a more comprehensive explanation of the algorithm and its theoretical underpinnings could help readers better grasp its contributions and the innovations it brings.
Thank You for pointing out this issue. We add explanations in Section 6 marked with yellow color – please see pp. 10-11 of the revised version of the manuscript.
- Literature Review and References:
While the paper presents an interesting concept, much of the referenced literature appears outdated, with most references exceeding 10 years in age. Incorporating more recent studies, particularly those related to 5G and B5G systems, would strengthen the relevance of the work and provide a more robust foundation for your claims.
We have reviewed new publications, especially dated 2003 and 2024, in MU-MIMO precoding area and CSI estimation area, we added new literature references (most valuable and recent ones) with explanations based on recent study and took into account publications, related to 5G and B5G systems. The following literature references were added ([27, 28, 35, 36, 37, 38], highlighted in yellow) in Introduction and other sections (please see pp. 2, 11, 17):
- R. Zhang et al., "Channel Estimation for Movable-Antenna MIMO Systems via Tensor Decomposition," in IEEE Wireless Communications Letters, vol. 13, no. 11, pp. 3089-3093, Nov. 2024, doi: 10.1109/LWC.2024.3450592;
- G. Chen, R. Zhang, H. Zhang, C. Miao, Y. Ma and W. Wu, "Energy-Efficient Beamforming for Downlink Multi-User Systems with Dynamic Metasurface Antennas," in IEEE Communications Letters, doi:10.1109/LCOMM.2024.3513554;
- J. -C. Chen and C. -H. Hsu, "Beamforming Design for Dynamic Metasurface Antennas-Based Massive Multiuser MISO Downlink Systems," in IEEE Open Journal of the Communications Society, vol. 5, pp. 1387-1398, 2024, doi:10.1109/OJCOMS.2024.3367984.
- R. M. Dreifuerst and R. W. Heath, "Machine Learning Codebook Design for Initial Access and CSI Type-II Feedback in Sub-6-GHz 5G NR," in IEEE Transactions on Wireless Communications, vol. 23, no. 6, pp. 6411-6424, June 2024, doi: 10.1109/TWC.2023.3331313.
- A. Hindy, U. Mittal and T. Brown, "CSI Feedback Overhead Reduction for 5G Massive MIMO Systems," 2020 10th Annual Computing and Communication Workshop and Conference (CCWC), Las Vegas, NV, USA, 2020, pp. 0116-0120, doi: 10.1109/CCWC47524.2020.9031236.
- A. Barannikov, I. Levitsky, V. Loginov and E. Khorov, "CSI Compression Method With Dual Differential Feedback for Next-Generation Wi-Fi Networks," in IEEE Wireless Communications Letters, doi: 10.1109/LWC.2024.3510215.4 4.Conclusion:
The conclusion could be more precise and aligned with the presented results. Currently, statements such as “The proposed precoding algorithm with quantization can be used for dynamic downlink user selection in MU-MIMO systems with a large number of antennas (5G and B5G)”feel somewhat ambiguous without clearer ties to your findings. Reiterating how the results validate the approach and explicitly connecting the outcomes to practical applications would make the conclusion more compelling.
These improvements would enhance the clarity, impact, and academic rigor of your paper.
Thank you for your contribution to this important area of research. I look forward to seeing the revised manuscript.
Thank You for these valuable comments and suggestions. The Conclusion section has been corrected and supplemented – please see pp. 16, 17.
We are looking forward to Your reply.
Kind Regards, authors.

Reviewer 3 Report
Comments and Suggestions for Authors
- As usual, a description is missing at the end of the introduction section, i.e., in Section ...
-. What is the main contribution achieved by this research? It is mentioned as an algorithm, however, it is not clearly explained compared to the literature review which leads one to think it is a sort of incremental contribution, or if it is enough contribution compared to existing literature, please reinforce all contributions in detail one by one.
-. What novelties are added in this work while considering the channel types, discuss more on this, slow, fast fading, Rician, Raileigh, etc..
- A notation section is missing as it is hard to follow sometimes the mathematical notations. Preferably just after the introduction section.
- How do the authors conclude with the evidence and arguments obtained from their numerical results? I believe this should be more elaborated in terms of more simulated case studies for examples while varying other parameters.
- The authors should update with more recent references some of them listed in this version. This would enrich the novelties proposed.
- In general, it looks like a high-level content paper, however, I believe that it can be further improved, thus up to now from my side I recommend minor revision before publication.
Author Response
Point-by-point response to the Reviewer's comments
Sensors Journal
Manuscript ID: sensors-3431905
Multi-User MIMO downlink precoding with dynamic users selection for limited feedback
Dear Editor and Reviewers,
Thank You very much for your time and valuable and constructive comments on our article. We took into account all Your comments and hope the quality of the paper was enhanced. In the following text we give our answers to the comments received point-by-point. For the convenience each response was marked in yellow color and we also highlighted the updates in yellow in the revised version of the article.
Corresponding authors,
Dr. Mikhail Bakulin, Dr. Taoufik Ben Rejeb, Prof. Dr. Vitaly Kreyndelin, Dr. Denis Pankratov and Dr. Aleksei Smirnov,
January 21, 2025
REVIEWER # 3
Thank You for these valuable comments, we took them into account in the revised version of the article (all updates and corrections are highlighted in yellow color). Below, for the convenience, we give the detailed answers point-by-point.
Open Review
(x) I would not like to sign my review report
( ) I would like to sign my review report
Quality of English Language
(x) The quality of English does not limit my understanding of the research.
( ) The English could be improved to more clearly express the research.
|
|
|
|
Yes |
Can be improved |
Must be improved |
Not applicable |
- As usual, a description is missing at the end of the introduction section, i.e., in Section ...
Thank You for pointing out this issue. We added this description at the end of Introduction section (marked in yellow color)
-. What is the main contribution achieved by this research? It is mentioned as an algorithm, however, it is not clearly explained compared to the literature review which leads one to think it is a sort of incremental contribution, or if it is enough contribution compared to existing literature, please reinforce all contributions in detail one by one.
Thank You for the valuable comments. We made corrections and additions in Introduction section. The purpose of the paper and explanation of how the proposed approach differs from known ones in the context of existing literature were added. Please see updated Introduction in the revised version of the manuscript. The Abstract has been modified too.
-. What novelties are added in this work while considering the channel types, discuss more on this, slow, fast fading, Rician, Raileigh, etc..
We have added clarifications regarding channel parameters - slow independent Rayleigh fading. This is added to the Table 1 with description of simulation parameters.
- A notation section is missing as it is hard to follow sometimes the mathematical notations. Preferably just after the introduction section.
Mathematical notations are introduced immediately after each corresponding equation. Special notation section is not necessary for our not large paper.
- How do the authors conclude with the evidence and arguments obtained from their numerical results? I believe this should be more elaborated in terms of more simulated case studies for examples while varying other parameters.
Thank You for the valuable suggestions. A simple precoding algorithm using only Grassmannian quantization (“Grassm. only”) does not work for this simulation case with 32 antennas at the base station and 32 mobile stations with one antenna each, since with such quantization there is not enough CSI information to provide efficient precoding for all 32 users. For such a number of users, it is necessary to use a Grassmannian with a significantly larger number of codewords, which is unacceptable due to the high computational complexity. We also added additional explanations on pp. 14-16 in the revised version of the paper.
- The authors should update with more recent references some of them listed in this version. This would enrich the novelties proposed.
Thank You for this comment. We have reviewed new publications and added new references (most valuable and recent ones) with explanations based on recent study. The following literature references were added ([27, 28, 35, 36, 37, 38], highlighted in yellow) in the manuscript:
- R. Zhang et al., "Channel Estimation for Movable-Antenna MIMO Systems via Tensor Decomposition," in IEEE Wireless Communications Letters, vol. 13, no. 11, pp. 3089-3093, Nov. 2024, doi: 10.1109/LWC.2024.3450592;
- G. Chen, R. Zhang, H. Zhang, C. Miao, Y. Ma and W. Wu, "Energy-Efficient Beamforming for Downlink Multi-User Systems with Dynamic Metasurface Antennas," in IEEE Communications Letters, doi:10.1109/LCOMM.2024.3513554;
- J. -C. Chen and C. -H. Hsu, "Beamforming Design for Dynamic Metasurface Antennas-Based Massive Multiuser MISO Downlink Systems," in IEEE Open Journal of the Communications Society, vol. 5, pp. 1387-1398, 2024, doi:10.1109/OJCOMS.2024.3367984.
- R. M. Dreifuerst and R. W. Heath, "Machine Learning Codebook Design for Initial Access and CSI Type-II Feedback in Sub-6-GHz 5G NR," in IEEE Transactions on Wireless Communications, vol. 23, no. 6, pp. 6411-6424, June 2024, doi: 10.1109/TWC.2023.3331313.
- A. Hindy, U. Mittal and T. Brown, "CSI Feedback Overhead Reduction for 5G Massive MIMO Systems," 2020 10th Annual Computing and Communication Workshop and Conference (CCWC), Las Vegas, NV, USA, 2020, pp. 0116-0120, doi: 10.1109/CCWC47524.2020.9031236.
- A. Barannikov, I. Levitsky, V. Loginov and E. Khorov, "CSI Compression Method With Dual Differential Feedback for Next-Generation Wi-Fi Networks," in IEEE Wireless Communications Letters, doi: 10.1109/LWC.2024.3510215.
- In general, it looks like a high-level content paper, however, I believe that it can be further improved, thus up to now from my side I recommend minor revision before publication.
Thank You very much for Your time, valuable comments and suggestions.
We took into account almost all of Your comments and looking forward to Your reply.
Kind Regards, authors.

Round 2
Reviewer 1 Report
Comments and Suggestions for Authors
Accept in current version
Author Response
Thank you very much for your time and efforts in reviewing the article.
Kind Regards,
authors
Reviewer 2 Report
Comments and Suggestions for Authors
Following my second review of the manuscript “Multi-User MIMO Downlink Precoding with Dynamic User Selection for Limited Feedback,” I am pleased to report that the authors have made substantial improvements in response to my initial feedback.
Key improvements observed include:
1. Introduction and Research Gap:
o The authors have significantly enhanced the articulation of the research gap, providing a clearer distinction of their proposed approach and its contributions.
2. Methodology:
o The methodology section has been improved with a more detailed explanation of the algorithm and its theoretical foundations, making the contributions more comprehensible.
3. Literature Review and References:
o The authors have addressed previous concerns by incorporating more recent references, ensuring the work is aligned with current developments in 5G and B5G systems.
Overall, these revisions have enhanced the manuscript’s clarity, impact, and academic rigor. Only minor language corrections are required before publication.
I appreciate the authors' efforts and believe the manuscript is now in a much-improved state.
Best regards,

Author Response
Dear Reviewer,
we are very grateful for Your efforts in reviewing the article and
for Your detailed comments and suggestions to improve the article, that we have tried to take into account.
Minor language corrections will be made in the latest version of the uploaded manuscript.
Kind regards,
authors